# Moisturizer in Patients with Inflammatory Skin Diseases

**DOI:** 10.3390/medicina58070888

**Published:** 2022-07-01

**Authors:** Seok-Young Kang, Ji-Young Um, Bo-Young Chung, So-Yeon Lee, Jin-Seo Park, Jin-Cheol Kim, Chun-Wook Park, Hye-One Kim

**Affiliations:** Department of Dermatology, Kangnam Sacred Heart Hospital, Hallym University, Seoul 24252, Korea; tjdjrdud@naver.com (S.-Y.K.); ujy0402@hanmail.net (J.-Y.U.); victoryby@naver.com (B.-Y.C.); minggijeook@gmail.com (S.-Y.L.); pahajs@gmail.com (J.-S.P.); aiekfne@naver.com (J.-C.K.); dermap@hanmail.net (C.-W.P.)

**Keywords:** cosmetics, moisturizer, cosmeceutical

## Abstract

As interest in skin increases, the cosmetic market is also growing. It is difficult to choose between the numerous types of basic cosmetics on the market. This article aims to provide advice and guidance on which products to recommend according to a patient’s skin condition. Appropriate application of a moisturizer attempts not only to improve the dryness, but also improve the skin’s natural barrier function to protect the skin from internal and external irritants to keep the skin healthy. Moisturizers consist of various ingredients, including occlusive agents, emollients, humectants, lipid mixture, emulsifiers, and preservatives. Pathophysiology of dry skin is also discussed to provide readers with the background they need to choose the right moisturizer for themselves. As moisturizers play an important role as adjuvant in the treatment of common skin diseases, such as atopic dermatitis, contact dermatitis, psoriasis, acne and rosacea, which type of moisturizer is appropriate for each disease was also dealt with. Basic cosmetics, especially moisturizers, should be recommended in consideration of the ingredients, effectiveness and safety of each product, and the skin condition of each patient.

## 1. Introduction

Cosmetics and pharmaceuticals are strictly separated in many countries, but many consumers expect physiological effects from their cosmetics similar to those of pharmaceuticals. The term that was born out of an attempt to express this idea is “cosmeceuticals” proposed by Dr. Albert Kligman [1]. The term “cosmeceutical” is a compound word of “cosmetic” and “pharmaceuticals,” and it considers cosmetic products with a certain degree of physiological activity as hybrid or intermediate between the two poles of cosmetics and drugs [2]. Even now, although cosmeceuticals are not recognized as a category by the Food and Drug Administration in almost all countries, including the US FDA, the term cosmeceutical is commonly used in various international symposiums and seminars, and is a term that is actually recognized by consumers in the global market [3]. According to Dr. Albert Kligman, to become a cosmeceutical, all of three conditions must be met: 1. Can the active ingredient penetrate the stratum corneum and can it be provided in a sufficient concentration to the skin through a process consistent with the mechanism of action? 2. Does the active ingredient have special biochemical and pharmacological mechanisms in the cells and tissues of human skin? 3. Is there a statistically significant double-blind placebo-controlled clinical study to substantiate the efficacy claim, and has it been published in a medical journal?

In this article, among cosmeceuticals, it is believed that moisturizing agents can serve as adjuvant treatments for atopic dermatitis, contact dermatitis, and psoriasis. Therefore, it is thought that doctors and medical experts can help in the treatment and research of skin diseases through a comprehensive review of moisturizers. The investigated role of moisturizers and the effects, safety, and types of the main ingredients included in moisturizers are reported. This was conducted to inform the proper selection of a moisturizer that performs a preventive and therapeutic role in various skin diseases accompanied by damage to the skin barrier or inflammation. The term basic cosmetic is a concept that includes detergents and moisturizers distributed in various forms such as lotions, creams, and gels. In this article, the selection criteria and proper use of basic cosmetics will be the focus, especially for patients with skin diseases.

## 2. The Role of Moisturizers in the Skin Barrier

The primary function of a moisturizer is to suppress moisture loss. When the skin barrier is damaged, it is necessary to first suppress the loss of water to restore the normal barrier [4]. A moisturizer supplies moisture to dry skin that is rough and flaky, which may be caused by dry environment and external stimuli [5]. In addition, a moisturizer helps to maintain and restore skin flexibility by blocking the evaporation of moisture, and induces uniform exfoliation of old dead skin cells to maintain a smooth skin surface [6]. When the moisturizer functions properly in the skin, the skin maintains homeostasis despite changes in the external environment, so that the stratum corneum can maintain proper hydration [7]. After applying a moisturizer to damaged skin, the skin barrier is restored through the four-step process that follows [8]. (1) The oily component of the moisturizer creates a thin film on the skin, and barrier repair begins; (2) The skin moisture distribution coefficient changes; (3) Moisture diffuses from the dermis to the epidermis; (4) Water distribution to the epidermis is controlled by the synthesis of skin lipids and intercellular lipid secretion.

That is, the oily component of the moisturizer forms a film on the skin to prevent the evaporation of moisture, the humectant component directly supplies moisture to the epidermis and dermis, and the emollient component controls the exfoliation of dead skin cells to make the skin barrier restored. In addition, when components similar to lipids in the stratum corneum of the skin are included in the moisturizer, the lipid component is supplied to the keratinocytes of the epidermis to rearrange the intercellular lipids [9,10], thereby maintaining the moisture content in the epidermis and improving the function of the skin barrier [11].

Appropriate application of a moisturizer not only improves the dryness, but also improves the skin’s natural barrier function to protect the skin from internal and external irritants to keep the skin healthy [12]. In addition, because various skin diseases accompany abnormalities in the skin barrier function, excellent moisturizing agents serve as adjuvants to treat and prevent diseases, and also to relieve skin disease symptoms such as itching and stinging [13].

## 3. Ingredients of Moisturizers

Moisturizers consist of occlusive agents that form an airtight film on the skin surface to suppress moisture loss, emollients that give a soft and smooth feeling, humectants that help the stratum corneum of the skin to contain moisture, lipid components (ceramides and other barrier lipids), and many other base components [14]. Occlusive agents, softeners, and lipid components are often oil-soluble, and humectants are often water-soluble. Most moisturizers are made by appropriately mixing these components. In addition to these base ingredients, emulsifying ingredients, preservatives, fragrances, and cosmetic drugs (active ingredients) with various functions may be included [15].

Many moisturizer ingredients serve more than one function. For example, if a substance with the properties of a sealant or softener has structurally hydrophilic and lipophilic functional groups, it also functions as an emulsifier. In October 2008, the entire ingredient labeling system was implemented in Korea, allowing consumers to check ingredient labels when purchasing cosmetics. Because all cosmetic ingredients are marked in the order of concentration contained in the product, it is possible to determine to some extent whether the ingredients and order of the moisturizer are centered on the sealant or the humectant [16]. Therefore, it is necessary to have a clear understanding of the classification and characteristics of these ingredients of moisturizers.

### 3.1. Occlusive Agents

Occlusive agents are usually oily substances that coat the stratum corneum (SC) rendering an emollient effect as well as the ability to decrease transepidermal water loss (TEWL) [17]. In general, the higher the oil component, the greater the softening action. The sealing and softening action is larger in the order of ointment, oil-based cream, water-based cream, and lotion.

The advantage of the occlusive agent is that it blocks moisture loss most effectively when applied immediately after washing or bathing. The disadvantage is that most are effective only when applied to the skin, and when removed from the skin through washing, the effect of inhibiting moisture loss disappears. If it is too strong, there is a possibility of bacterial growth in the stratum corneum (SC). Therefore, when using an occlusive agent as moisturizer, the degree of sealing should not lower the rate of moisture loss through the epidermis by 40% or less; thus, it is often mixed with a humectant [18].

Among the currently available occlusive agents, the components with the best sealing performance are Petrolatum, which is a semi-solid form of petroleum refined products, and mineral oil, which is a liquid form. Here we will examine components of occlusive agents such as hydrocarbons which include petroleum, silicone, vegetable oils and vegetable fat, waxes, and fatty acids and fatty alcohols.

#### 3.1.1. Hydrocarbons (Petrolatum and Mineral Oil)

Hydrocarbons are compounds made of carbon and hydrogen. They are chemically inert substances that do not have polar groups.

##### Petrolatum

Petrolatum is a hydrocarbon extracted from petroleum that was patented and brought into use in the United States in 1872 by Robert A. Chesebrough. Its brand name was “Vaseline,”; a combination of the Saxon word wassor (water) and the Greek word oleon (oil) [19]. Other names include petroleum jelly, white petrolatum, and soft paraffin. It is known to exhibit no rancidity and induce no comedones as it is tasteless, odorless, and neutral, does not cause allergy, and provokes little skin irritation because, except for carbon and hydrogen, other aromatic and unsaturated structures are removed during the purification process. Since the 1880s, it has been one of the most used raw materials in cosmetics and is an excellent ointment base used to formulate most drugs. Petrolatum is an excellent occlusive agent that reduces moisture loss through the epidermis by 99% [20] and is 170 times more effective in inhibiting moisture loss than olive oil, a vegetable oil. However, when applied to the skin, the feeling of stickiness is severe [21]. Thus, its feeling in use is not good and it is shiny. For these reasons, it is generally used for such as lip protectant (lip balm) or a diaper rash ointment, rather than for facial uses. 

##### Mineral Oil

Similar to Petrolatum, mineral oil is extracted from crude oil and is called mineral oil to contrast with vegetable oil. Other names include white oil, liquid paraffin, paraffinum liquidum, and liquid petroleum. Mineral oil is colorless and odorless, and is a component commonly used as an occlusive agent to the extent that a product with added fragrance has been commercialized as baby oil. Compared to petrolatum, the degree of reduction in transdermal moisture loss is less when applied, but it has less stickiness and better spreadability. Mineral oil is controversial in terms of induction of comedones, but one study concluded that well-refined mineral oil did not induce comedones [22].

#### 3.1.2. Silicone

Silicone is a compound of the silicon present in sand or quartz. A very strong bond called a siloxane bond (-Si-O-Si-) is formed between silicon and oxygen, resulting in stability to heat, ultraviolet light, acid, alkali, electrical stimulation, and oxidation reaction. Silicone does not dissolve in water, but it can emit water vapor, so it can be used in cosmetics with a waterproof function without worrying about miliaria. Because it does not mix with the sebum of the skin as well as water, the cosmetics are not easily erased by the sebum. Silicone has good spreadability, is not sticky, induces fewer comedones and allergies, and has no irritating scent. Silicones are commonly used ingredients in products labeled as “oil-free.” Dimethicone and cyclomethicone are commonly used. Dimethicone has a moisturizing effect as well as a softening effect, so it softens the dead skin cells of the skin and can vary the viscosity, increasing the stability of emulsified cosmetics such as creams and lotions [23]. Dimethicone is the second most common active agent in moisturizers today, following petrolatum, due to its hypoallergenic, noncomedogenic and nonacnegenic properties [24]. Toxicity due to topical application of silicone components has not been reported yet.

#### 3.1.3. Vegetable Oils and Vegetable Fat

Oils and fats obtained by pressing various plants or extracting solvents are called vegetable oils, which are liquid at room temperature, and vegetable fats, which are solid. Vegetable oils include castor oil, olive oil, camellia oil, macadamia nut oil, and grape seed oil, and vegetable fats include cocoa butter and shea butter. They are mainly composed of triglycerides consisting of fatty acids and glycerin in their chemical structure. However, they also contain unsaturated essential fatty acids of 18 carbons, such as oleic acid and linoleic acid. These unsaturated fatty acids also play a good role in the barrier function, such as acting as a PPAR alpha agonist [25], but when used as a cosmetic, they oxidize readily and cause a smell, so antioxidants must be added. According to the recent consumer preference for natural raw materials, each brand claims that various vegetable oils are added to enhance the moisturizing function [26].

#### 3.1.4. Waxes

Wax is an ester composed of a fatty acid and an alcohol in terms of chemical structure, and generally has a high melting point [27]. 

##### Beeswax

Beeswax is secreted by bees when they make a honeycomb, and it has a very chemically stable structure [28]. It is mainly composed of free cerotic acid and myricyl palmitate. It is used in moisturizers used on the lips such as lip balms, lipsticks, and lip glosses, as well as hair waxes and sunscreens.

##### Lanolin

Lanolin is an oil produced by the sebaceous glands of sheep and which coats their wool [29]. Lanolin, also called wool fat, is a wax without glycerides. The main component of lanolin is a long-chain wax ester. Refined lanolin used in pharmaceuticals and cosmetics is a pale-yellow ointment-like substance and is an odorless substance containing fatty acids and alcohols in a one-to-one ratio [29]. Lanolin is water-soluble and becomes a useful emulsifying agent when mixed with other lipids. Since lanolin is a common allergen, lanolin extracts or derivatives should be used instead of natural lanolin when used as occlusive agents [29].

#### 3.1.5. Fatty Acids and Fatty Alcohols

Fatty acids are esters (R-COOH), the main component of animal oils and fats, and most of them are distilled by hydrolyzing triglycerides [30]. Long-chain fatty acids, such as palmitic acid and stearic acid, are commonly used. In particular, stearic acids are widely used in cosmetics as emollients and emulsifiers because they are saturated fatty acids, liquid, and chemically or microbially stable [31]. Alcohols used in cosmetics include cetyl alcohol and stearyl alcohol, which are also used in creams or emulsions as emulsifiers, as well as softening agents.

### 3.2. Emollients

Emollient refers to an ingredient that fills the gaps in the stratum corneum and gives a soft feeling. On dry skin, the keratin mass is removed leading to a rough feeling, so an emollient component is added in addition to the moisturizing component. Although many emollient ingredients have moisturizing functions, there are also emollients that do not reduce transepidermal moisture loss; that is, they have little moisturizing function. Emollients can be classified as protective, fatting, astringent, or dry according to their inherent properties [32]. Emollients include a variety of high-grade alcohols and esters. Among them, high-grade alcohols such as cetyl alcohol and stearyl alcohol used as emollients do not dry the skin (unlike isopropyl alcohol and ethyl alcohol, which act as astringents) and give a soft feeling when applied to the skin [33]. Ester type emollients include octyl stearate, isopropyl myristate, oleyl oleate, cetearyl isononanoate, and PEG-7 glyceryl cocoate. In addition, lanolin, mineral oil, and petrolatum are substances that act as an emollient and occlusive agent at the same time [8]. Substances that act as emollients are listed in Table 1 [8].

### 3.3. Humectants

Humectant refers to substances which, when applied to the skin surface, pull both moisture in the atmosphere and moisture below the stratum corneum toward the stratum corneum. When the humidity in the atmosphere is less than 80%, it mainly functions to attract moisture below the stratum corneum [34]. Most humectants have a molecular size of 200–500 kDa and can be absorbed into the stratum corneum. If the humectant is of small size (MW = 200–300 Da), such as glycerol or urea, deeper layers of SC are reached where they restore water content and barrier function, and replicate NMF function. Larger molecules (e.g., hyaluronic acid) do not penetrate the skin but increase the hydration of the outermost corneocytes [10]. They are substances that contain many hydroxyl (-OH) or amine (-NH) groups that can hydrogen bond with water molecules. Among the humectants mainly used in cosmetics, water-soluble polyalcohols (glycerin, -glycol) are the most common, and urea, lactate, pyrrolidine carboxylic acid (PCA), alpha hydroxy acid (AHA), polypeptide, hyaluronic acid, sorbitol, collagen, and elastin are also used. Among them, the main humectants are listed in Table 2 [8]. Because transepidermal moisture loss can increase when a humectant is applied alone, humectants are usually used together with an occlusive agent. 

#### 3.3.1. Glycerin 

Glycerin is a powerful humectant and has similar hygroscopicity to natural moisturizing factors. After topical application, it increases the moisture content inside and outside the keratinocytes, and prevents the lamellar structure of intercellular lipids from being transformed from plate to crystal [35]. It also helps to protect the skin barrier by regulating the expression of aquaporin-3, which is the primary aquaporin in the epidermis [24]. This effect is maintained to some extent even after the glycerin is removed from the skin surface. In addition, glycerin normalizes the enzyme activity that induces the decomposition of the keratinocyte complex so that the keratin is removed normally. 

#### 3.3.2. Pyrrolidine Carboxylic Acid (PCA)

PCA, as a component of natural moisturizing factor (12%), in cosmetics, has an optimal moisturizing effect at a concentration of 4% [36,37]. 

#### 3.3.3. Urea 

Urea is a component of a natural moisturizing factor (7%) and is mainly used as a hand cream. When the concentration is less than 10%, it has a moisturizing effect, and when it is higher than 10%, it shows a keratolytic effect. Urea is used as a 10% cream for the treatment of ichthyosis and hyperkeratotic skin disorders, and in lower concentrations for the treatment of less severe dryness [38,39,40]. However, it may sting even at a concentration of less than 10%, so the face and sensitive areas should be avoided, and it is not a suitable moisturizer for children. 

#### 3.3.4. Hydroxy Acids 

Alpha-hydroxy acids (AHAs) are a kind of natural organic acid that not only wets the stratum corneum of the skin but also decomposes the keratinocytes to exfoliate the dead skin cells [41]. As a result, the keratin becomes thinner, increasing the flexibility of the skin, and reducing the scales on the skin surface to give a smooth feeling. Glycolic acid and lactic acid are the most commonly used. Lactic acid is an AHA and a component of a natural moisturizing factor. It induces the formation of ceramides in keratinocytes as well as the exfoliation of dead skin cells. The concentrations used for treatment of dry skin disorders have ranged up to 12% [42], and the permitted use concentration for cosmetics is 5–10%. 

#### 3.3.5. Propylene Glycol 

Propylene glycol has both a hydrating function (at concentration 10% or less) and keratolytic function (at concentrations 40% or more) [43]. 

### 3.4. Other Ingredients 

#### 3.4.1. Emulsifiers

Moisturizers are a mixture of oil and water, which do not mix well with each other. An emulsion is a two-phase system made up of two immiscible components in which the dispersed phase is contained in the form of droplets within a continuous phase [44]. At this time, one phase is oil (oil phase) and the other phase is water (water phase). The water-in-oil (w/o) system disperses the aqueous phase in the oil phase to make oily cream or cold cream. The oil-in-water (o/w) system disperses the oil phase in the aqueous phase, so it makes a water-based cream or a vanishing cream that feels as though it disappears quickly when applied to the skin. Sometimes it is possible to create an emulsion with both forms in one system, which is called “ambiphilic cream” [45].

To make a stable emulsion by mixing the two components, a surfactant, that is, an emulsifying component, must be added. The emulsifier has a polar group with strong hydrophilicity and a non-polar group with strong lipophilicity at the same time, so it is a material with high molecular weight that connects across the gap between polar and non-polar substances. Surfactants can be classified into cationic, anionic, zwitterionic, and nonionic surfactants according to their ionization when dissolved in water. Among them, amphoteric surfactants or nonionic surfactants are mainly used for the purpose of emulsifying moisturizers in the form of lotions and creams. Amphoteric surfactants include amino acids, betaine-type, synthetic raw materials for imidazoline derivatives, and natural products such as lecithin [46]. Nonionic surfactants include a wide variety of substances, among which are hydroxyl groups (-OH), ether bonds (-O-), (-CONH-), and ester bonds (-COOR). Among nonionic surfactants, stearate, PEG (polyethylene glycol), and cetearyl alcohol are commonly used as moisturizing agents.

#### 3.4.2. Preservatives

Moisturizers are stored at room temperature for a considerable period of time after opening, and people frequently touch the opening of the container with hands so there is a high probability of contamination. In general, moisturizers with low moisture such as ointments and oil-based creams do not require preservatives, but lotions, o/w creams, and gels contain a large amount of moisture so if they are contaminated by bacteria or fungi, there is a high probability of microbial growth, so it is necessary to use preservatives [47]. To prevent the growth of these microbes, various preservatives or substances that discharge preservative components are added to the moisturizer. Parabens, phenoxyethanol, sorbic acid, propylene glycol, thimerosal, ethylene diamine-tetraacetate (among others) are used as preservatives [48]. Antioxidants such as gallate, butylhydroxyanisole (BHA), and butylhydroxytoluene (BHT) are used to prevent the rancidity of unsaturated fats in oil-based moisturizers and lipsticks [49].

## 4. Moisturizers for Repairing the Barrier Function

Two of the functions of the lipid matrix in the stratum corneum are (1) to prevent excessive water loss through the epidermis and (2) to avoid that compounds from the environment permeate into the viable epidermal and dermal alyers and thereby provoke an immune response [50]. Ceramide, cholesterol, and fatty acids, which are the main components of keratinocyte interstitial lipids, have been found to be required for permeability barrier homeostasis [51]. In this context, moisturizers containing these components are being developed. In particular, the concept of a physiological lipid mixture in which ceramide or pseudoceramide, cholesterol, and free fatty acids are mixed in an ideal ratio (1:1:1 or 3:1:1) of stratum corneum lipids is applied to moisturizers. When applied to the skin, normal lipids act as sealants or softeners in the stratum corneum, whereas the physiological lipid mixture passes through the stratum corneum. Physiological lipids are also used for lipid synthesis in granular keratinocytes, are stored in the keratinocytes, and are transported between keratinocytes, directly participating in restoration of the skin barrier. Therefore, when the physiological lipid mixture and the occlusive agents are applied together, the occlusive agent prevents moisture loss within a short period of time, and the lipid mixture gradually restores the skin barrier function, thereby recovering dry skin. Based on this concept, several products have been developed as moisturizing agents for atopic dermatitis with impaired barrier function. Although there is controversy over the ability of the epidermis to synthesize lipids when the application is stopped after applying a moisturizer containing this physiological lipid mixture for a long time, it is known that most moisturizers do not reduce the synthesis of epidermal lipids [52,53]. 

## 5. Classification of Moisturizers

Many cosmetic companies are selling not only moisturizers distributed through hospitals and pharmacies, but also a variety of moisturizers through diverse other distribution channels. In addition to products that claim to increase the moisture content of the skin, almost all basic cosmetics, including lotions, creams, gels, and ointments, contain base ingredients that have a moisturizing function. Thus, in fact, almost all basic cosmetics can be said to be included in the moisturizer category. These basic cosmetics that act as moisturizers are divided according to their popularly sold formulations [54], as shown in Table 3.

## 6. Pathophysiology of Dry Skin

Healthy stratum corneum (SC) contains from 15% to 25% water at the skin surface, and to about 40% at the SC/stratum granulosum border [55]. When the skin barrier structure is damaged, dry skin occurs in which the water-holding capacity of the stratum corneum is reduced. When dry skin is subdivided according to the Baumann Skin Typing System, dry skin is divided into three stages: dry, slightly dry, and extremely dry [56]. Dry skin conditions can be further classified as combination skin and sensitive skin. Considering that dry skin increases with age, the role of a moisturizer is likely to play the most basic skin care role. Healthy skin should generally be able to store 10–15% water [57]. Too high stratum corneum moisture, such as when your hands are soaked in water for a long time, or too little moisture impairs the barrier function.

Dry skin is a very common condition that most people experience at some point in their lives. The use of moisturizing products, popularly called emollients or moisturizers, is the key to treating dry skin. Because the causes, symptoms, and severity of dry skin are very diverse, the choice of moisturizer varies from person to person. Dry skin may not have any symptoms to complain about, depending on the degree of subjective dryness, but many may complain of a feeling of pulling or tightening, stinging, or cutting, especially if it is accompanied by other diseases. Depending on the degree of dry skin, various clinical findings are shown, including such as roughness, dull skin tone, scaling, redness, cracks, and fissures. Several factors are involved, including both exogenous (e.g., climate, environment, lifestyle) and endogenous (e.g., medication, hormone fluctuations, organ diseases) factors [57]. There are also endogenous factors such as loss of barrier lipids, lack of natural moisturizing factors (NMF), abnormal moisture delivery, and abnormality of the keratin exfoliation cycle.

### 6.1. Abnormalities of Intercellular Lipids of Stratum Corneum

The stratum corneum lipids surround the keratinocytes, which contain adequate moisture through natural moisturizing factors, and prevent the moisture of the skin from easily evaporating. The stratum corneum lipids account for about 15% of the dry weight of the stratum corneum, among which ceramides (50%), cholesterol (25%), and fatty acids (20%) are particularly related to the barrier function [58]. The stratum corneum lipids are well arranged in multiple layers around the lipid envelope bound to the cornified cell envelope, and serve as an epidermal barrier. That is, stratum corneum lipids play a role in preventing water loss through the epidermis, and conversely, prevent water-soluble substances from passing through the epidermis and entering the human body. In truly dry skin, the multilayer structure of the lipid membrane is damaged, and the fatty acid content is increased while the ceramide content is lowered [59]. This damage to the lipid membrane results in an increase in transepidermal water loss. A pH that is too high promotes the degradation of the stratum corneum lipid membrane. Damage to the lipid membrane can be caused by various causes, such as ultraviolet rays, use of detergents including surfactants, acetone, chlorine bleach, friction, and excessive hydration.

### 6.2. Lack of Natural Moisturizing Factors (NMF)

A natural moisturizing factor is a kind of humectant that exists inside keratinocytes and plays a role in adequately hydrating the stratum corneum [60]. Many researchers have found a link between a reduction in the levels of NMF and dry skin [61]. The NMF is found within corneocytes and is a mix of hygroscopic molecules which keep the SC hydrated, helping maintain hydration in the corneocyte [62]. When a protein called filaggrin, which plays a role in arranging and bonding keratinocytes in keratinocytes is decomposed, urocanic acid, pyrrolidone carboxylic acid (PCA), and several amino acids are produced. These osmotic amino acids and other natural moisturizing factors exist in keratinocytes and have a moisturizing function, so they act as a factor to maintain moisture in the skin. Genetic alterations in filaggrin metabolism are associated with impaired barrier function and reduced water binding capacity and are the etiology of certain pruritus and atopic dermatitis cases [63]. Other natural moisturizing factors include lactic acid, urea, and inorganic ions such as sodium, potassium, calcium, and chloride, all of which help to maintain the epidermis’ moisture content.

### 6.3. Abnormal Desquamation of the Stratum Corneum

A normal desquamation process occurs in very small units of keratinocytes and is invisible to our eyes. Clinically dry skin appears to be covered with scales, which is caused by abnormal keratinocyte exfoliation. An important factor controlling the exfoliation of the stratum corneum is the degree of activation of various proteolytic enzymes, which are controlled by the pH and hydration level of the skin [64]. In dry skin caused by winter or excessive use of soap, the corneodesmosome of the upper stratum corneum is increased compared to normal skin and desmoglein 1 is increased in the surface stratum corneum. When the moisture content is low, the function of the enzyme that removes them is reduced, making skin look rough [65]. The aggregated dry keratinocytes are visible. Conversely, when the stratum corneum is hydrated, a lacuna is formed in a hydrophilic keratinocyte site due to the action of a proteolytic enzyme, and keratin exfoliation occurs. As this series of unnatural processes occurs, the phenomenon of keratin aggregation in the form of visible scales happens, after which they fall off at once.

## 7. Basic Principles for Using Moisturizers on Dry Skin

Basic skincare for the treatment of dry skin is intended to improve skin moisturization, supplement barrier lipid deficiency, and improve skin barrier function. Therefore, it is basic to use moisturizers with a good combination of hydrophilic and hydrophobic ingredients, and products with low fragrance and allergens are recommended. The lower the lipid content of the stratum corneum, the better the local fat-rich agents can penetrate, and in extreme dry cases, moisturizers containing more lipids in the base can quickly relieve dry symptoms by suppressing moisture loss [66].

It is recommended that dry skin be cleansed and showered with lukewarm water and a synthetic cleanser (Syndet) made of mild surfactant be used that does less damage to skin lipids and natural moisturizers and that users refrain from using regular soaps [67]. The ratio of surfactants that can irritate the skin is reduced in the order of cationic > anionic > amphoteric > non-ionic surfactants. In addition, the larger the size of the micelle of the surfactant, the less it penetrates the stratum corneum, so it is recommended to refer to the selection of cleaning agents for dry skin. It is recommended to avoid using alkaline soap because dry skin has impaired barrier function and degraded pH recovery.

## 8. Selection of Basic Cosmetics for Individual Skin Diseases

Moisturizers are used not only for the purpose of normal skin care, but also for the treatment of skin diseases with reduced skin barrier function or for clinically dry skin. In particular, moisturizers play an important role as adjuvant in the treatment of common skin diseases such as atopic dermatitis, contact dermatitis, psoriasis, acne, and rosacea.

### 8.1. Atopic Dermatitis

Moisturizers that can restore the damaged barrier function, the main pathophysiology of atopic dermatitis, are not just cosmetics, but also act as therapeutic agents. As research on the skin barrier progresses, moisturizers containing natural moisturizing factors, ceramides, and pseudoceramides have been released under the concept of a therapeutic moisturizer, and are widely used by patients with atopic dermatitis.

Prof. Pester Elias’s team investigated the effects of the external application of three types of lipids (ceramide, cholesterol, and free fatty acids) that constitute intercellular lipids [45]. When applied alone or in a mixture of two, the treatment did not help to restore the barrier function but rather weakened it due to the direct toxic effect of the lipid. However, when the three types were mixed in an appropriate ratio and applied, the restoration of the barrier function was accelerated. Based on the results of these studies, the concept of a physiological lipid mixture showing optimal synergistic effect was proposed. The mixing ratio of ceramide, cholesterol and free fatty acids is known to be most effective at a molar ratio of 3:1:1, and each concentration of 1–2% or more is appropriate. Moisturizers based on these research results went through clinical trials directly applied to patients and were approved by the US FDA as a therapeutic material for improving the symptoms of atopic dermatitis.

In addition, moisturizers for atopic dermatitis often contain natural moisturizing factors, such as urea, lactate, sodium PCA, and hyaluronate. Anti-inflammatory ingredients such as glycyrrhetinic acid and palmitoylethanolamide (PEA), along with various plant extracts with anti-inflammatory and antibacterial effects (such as Peony extract) are widely used [68]. However, although anti-inflammatory and antibacterial effects of individual ingredients are confirmed, because each ingredient is added in a small amount, there is a lack of evidence to prove the beneficial effects of cosmetic formulations in atopic dermatitis. The results of clinical studies on the use of cosmetics that claim to be effective for atopic dermatitis vary, ranging from just a slight relief of dry skin to an effect corresponding to that with steroid ointment, as well as increased antimicrobial peptides in the lesions. Because atopic dermatitis is a chronic disease, the selection of a moisturizer is important, but it is more important for patients to regularly use an appropriate amount of moisturizer through identification of the patient’s characteristics and the education of patients and caregivers.

In addition to moisturizers, cleaning agents are important skin care products that can help patients with atopic dermatitis. Two effects can be expected through proper bathing in atopic dermatitis patients: the action of cleaning away harmful irritants or allergy-causing substances and various pathogens (bacteria/viruses) present on the skin surface, and temporary promotion of skin hydration. Despite these positive effects, excessive bathing and use of strong detergents can destroy the skin’s natural moisturizing factor and lipid membrane, so the skin barrier function may be impaired. There are cases where patients and caregivers think only of these effects and do not bathe often, or even only with water. Alternatively, there is the opposite case of excessively washing with a detergent such as solid soap, which is strongly irritating to the skin. In order to enhance the effectiveness of the bath, it is necessary to use an appropriate cleaning agent. Although any type of cleaning agent used for bathing can be used as long as it does not significantly affect the stratum corneum acidity, a mild surfactant should be used if possible, and a mildly acidic, hypoallergenic detergent is recommended. Most synthetic detergents that come in the form of liquid, gel, or foam rather than solid soap, reflect these characteristics well. It is appropriate to shower or bathe once a day or every two days.

Meanwhile, there is controversy about the claim that applying a moisturizer prevents the occurrence of atopic dermatitis. Several studies showed that when emollient was applied daily to infants starting within 3 weeks of age, the relative risk of developing atopic dermatitis was reduced at 6 to 24 months of age [69,70]. On the other hand, other studies claimed that applying emollients showed reduction in AD incidence up to 12 months of age no benefit at 24 months of age, hypothesizing that the emollients delay the onset of AD rather than prevent it [71].

However, in other studies of similar design, moisturizer application did not reduce the incidence of AD [72]. A large randomized controlled trial conducted in 2020 reported that daily emollient application had no effect in preventing eczema in high-risk children, but could increase the risk of skin infection or disturb the natural protective skin microbiome [73].

Various factors can affect the effectiveness of the emollient, such as the type of emollient, the number of times of application, and the start time, so it is difficult to conclude yet. A new large randomized controlled trial, Prevention of Eczema by a Barrier Lipid Equilibrium Strategy (PEBBLES), which controlled for these factors, is currently underway [7,74].

### 8.2. Contact Dermatitis

Irritant contact dermatitis is dermatitis caused by acute or chronic damage to the stratum corneum of the skin due to exposure of the skin to an irritant substance, thereby damaging the barrier function. Although irritant contact dermatitis patients use moisturizers empirically, some moisturizers do not play a role in improving the barrier function other than a temporary softening effect on the skin, so the selection of a moisturizer with an appropriate proportion of occlusive agent and humectant is important. Regular use of a moisturizing agent with a high lipid content is effective in preventing irritant contact dermatitis caused by surfactants in normal skin and in treating existing irritant contact dermatitis [13,75].

The non-lesional skin of patients with allergic contact dermatitis has a slow barrier recovery rate and a significant decrease in ceramide content, showing a barrier defect as in atopic dermatitis [76]. The principle of using a moisturizer or cleaning agent for allergic contact dermatitis is similar to that of atopic dermatitis patients.

### 8.3. Psoriasis

Psoriatic lesions show a higher TEWL value compared to that of uninvolved sites [77]. The use of moisturizers for psoriasis is effective when supplemented with other treatments, such as steroids and vitamin D preparations. Moreover, the use of moisturizers such as mineral oil before UV treatment can reduce the reflectivity of dead cells in thick skin. In a randomized controlled trial in palmoplantar psoriatic patients, the degree of desquamation, surface area, and subjective symptom improvement was significantly greater when emollient was used together compared to when only steroids were used [41]. In another study, it is claimed that applying a moisturizer in a stable state while applying a steroid ointment intermittently for mild psoriasis prevents recurrence [78]. To relieve hyperkeratosis of psoriasis, one can use a moisturizer containing such as AHA, salicylic acid (BHA), urea, and glycolic acid, which have keratolytic action.

## 9. Moisturizer Selection Criteria, Usage, and Side Effects

### 9.1. Moisturizer Selection Criteria

A good moisturizer should be applied gently to the skin, retain moisture for a long time, and be able to restore the skin’s normal barrier function. The choice of moisturizer depends on the user’s skin type (i.e., dry, oily, or combination), the season, and the area to be treated. Moisturizer formulations are divided into lotions and cream ointments in the order of high moisture content. When choosing a formulation, skin type is the most important factor (dry, oily, normal, and combination are the selection criteria). For dry skin, select an ointment or oily cream formulation, and for oily skin, select a lotion and water-based cream formulation. The body part to be used is also a consideration. For hands, feet, shins, and lips, which are prone to dry skin due to their small amount of sebaceous glands, an ointment or cream formulation containing a large amount of occlusive agents should be selected. In addition, it is preferable to use a moisturizer-oriented product in summer and an oil-based cream formulation containing large amounts of occlusive agents in winter.

In skin diseases with accompanying dryness, the importance of moisturizers as an adjuvant therapy beyond simply moisturizing is being emphasized. In the past, moisturizers gave the skin the wetting function of natural moisturizing factors and the sealing function of sebum. Recently, in terms of restoration and strengthening of the skin barrier function, it is known that inter-keratinocyte lipid components such as ceramide and lamellar structure play a very important role in maintaining the skin barrier function and maintaining skin homeostasis. The ideal moisturizer should have the wetting function of natural moisturizing factors, the sealing function of the skin lipid membrane, and the barrier function of lipid between keratinocytes. This will be a product that can effectively restore the lamellar structure. Several domestic and foreign companies are highlighting the advantage of having their own lipid layered structure be well-fused to the stratum corneum lipid membrane. Many moisturizers use synthetic ceramides or similar pseudoceramides, or some have ceramide precursors or enhancers that can increase synthesis [79]. Moisturizers in the form of physiological lipid compounds are known to reduce damage to the skin barrier function caused by external steroids, so their use can be expected to increase as a therapeutic adjuvant.

### 9.2. How to Use Moisturizer

Moisturizer should be applied gently after wiping away the moisture on the skin after washing the face or bathing. Apply several times for dry skin, and reduce the number of times for oily skin. Moisturizers centered on humectants can evaporate epidermal moisture when exposed to cold, dry air after being applied to the skin, so it is recommended to apply the product 30 min before exposure to cold, dry air. If the skin will be exposed to dry air immediately after application, select a product that contains an occlusive agent. It is good to suppress water loss immediately.

### 9.3. Side Effects of Moisturizers

Side effects may occur when using cosmetics containing moisturizers. The stability of cosmetic ingredients is being treated as more and more important, and the cumulative amount of cosmetics that are frequently applied to the whole body, such as moisturizers, is considerable. It is especially important to choose products with safe ingredients and at safe concentrations because the absorption of cosmetic ingredients can increase if users have damaged skin barriers due to something comparable to atopic dermatitis, or because their skin is immature (infants and children).

When using cosmetics, the main side effects are irritating reactions such as stinging, itching, burning, and pulling. Moreover, symptoms of allergic contact dermatitis, photo contact dermatitis, and contact moles may occur. The main causative substances are preservatives, fragrances, and other additives, and their concentrations are important indicators as well as types. For example, propylene glycol, which is a moisturizer and serves as a preservative, is known to irritate frequently if the concentration is higher than 20%, but concentrations of 2 to 3% are known to cause little irritation [80]. In Korea, from 1 January 2020, the labeling of allergens in cosmetics and fragrances became mandatory. In addition, 25 types of allergen-inducing ingredients among the ingredients of fragrances determined by the Ministry of Food and Drug Safety are indicated by their ingredient names [81]. There are 25 allergens, including Alpha-isomethylionone, Amylcinnamal, Amylcinnamyl alcohol, Anisethanol, Benzyl alcohol, Benzylbenzoate, Benzylcinnamate, Benzyl salicylate, Butylphenylmethylpropional, Cinnamyl, Cinnamyl alcohol, Citral, Citronellol, Coumarin, Eugenol, Gerani All, Hexylcinnamal, Hydroxycitronellal, Isoeugenol, Limonene, Linalol, Methyl2-octinoate, Oaktree moss extracts, and Panesol.

When allergic or irritant reactions caused by moisturizing ingredients are suspected, a single or repeated patch test and open patch test are performed for diagnosis [64]. Because other long-term safety issues regarding the effect on reproductive function or the possibility of carcinogenesis are emerging in products such as preservatives and fungicides, the dermatologist should know the exact basis for any issue.

## 10. Conclusions

As the interest in beauty continues to increase, the use and demand for cosmetics in women, men, and children are increasing explosively. Numerous cosmetics are being released by a variety of cosmetic companies today, and in particular, moisturizers are used not only for patients suffering from skin diseases but also for people with normal skin. Moreover, it is often difficult to choose from among the numerous moisturizers available. With the information provided in this paper, it should be possible to compare and select moisturizers based on accurate scientific information for the effectiveness and safety of the bases and additives included in these moisturizers.

## Figures and Tables

**Table 1 medicina-58-00888-t001:** Emollient substances.

Alcohols	Esters
Octyldodecanol	Oleyl oleate
Hexyldecanol	Octyl stearate
Oleylalcohol	PEG-7 glyceryl cocoate Coco caprylate/caprate Myristyl myriate Cetyaryl isononanoate Isopropyl myriate

**Table 2 medicina-58-00888-t002:** Component of skin and moisturizer—hydrophilic humectant.

SC Component	Biomimetic Moisturizer Ingredient
Natural moisturizing factors	
Amino acid	Amino acid
Pyrrolidone carboxylic acid	PCA & salts
Lactate	AHA & salts
Urea	Urea
Others	Polyhydric alcohols: Glycerin, propylene glycol, sorbitol

**Table 3 medicina-58-00888-t003:** Classification of moisturizers.

	Type	Characteristics	Functions	Ingredient
Beauty wash	Flexible lotion, convergence lotion, cleansing lotion	Solubilizing the insoluble material in water soluble or micro-emulsion techniques make liquid cosmetic to show a state that is transparent ornon-transparent.	Used after cosmetic washing and restoring the skin surface pH to slightly acidic and supplying water.	Purified water (30 to 95%), alcohol (0.40), wetting agents (to 20%), the emollient agent, an emulsifier, detergent, solubilizing agent (surfactant, thickening agents), pH adjusting agents, perfumes, preservatives, pigments, discoloration inhibitors, biologically active substance
Lotion	o/w, w/o, w/o/w	Suitable for normal and oily skin during the summer because the feeling is neat due to the high proportion of water compared with oil.	Skin moisturizing and improving flexibility.	Similar to the ratio of the cream but the oil or wax components are low. Nonionic, anionic, amino acid-based surfactants are used for easier dissolving.
Cream	High/Low/Mid oil cream (o/w, w/o) Multiple cream (o/w/o, w/o/w)	Types of emulsions which disperse the two insoluble materials in a stable state Typically, higher proportion of such as humectants and oil than lotion.	Skin moisturizing, improving flexibility, supplying pharmaceutical ingredients for cosmetics.	Stearic acid, alcohol, water, stearic acid monoglyceride, etc.
Gel	Water type Oil type	Feeling very neat and possible to use a water-soluble drug.	Providing a cooling sensation on the skin.	▪ Water type Water-soluble polymer in gel (carboxyvinyl polymer, methyl cellulose), and other moisturizing agents, surfactants, a large amount of water and alcohol, etc. ▪ Oil type Complex of oil and surfactants in gel.
Essence	Wrinkles	The concentration of unparalleled beauty ingredient for skin.	Anti-wrinkles, moisturizing, softening effect, whitening, anti-acne effect	Water, lotion, cream, gel, etc.
Pack	Wiping Type Detaching type Attaching type	Improving blood circulation to seal the skin for a period of time. Making cosmetic ingredients that are contained in cosmetics to be well absorbed and hydrate the stratum corneum.	Moisturizing, flexibility, occlusive effect, cleansing effect, promotion of skin blood circulation	Gels, pastes, purified water, alcohol, humectants, thickeners, emollient agent, surfactant, etc.

## Data Availability

Not applicable.

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
