# Peer review of "Moisturizer in Patients with Inflammatory Skin Diseases"

_medicina, 2022, doi:10.3390/medicina58070888_

Round 1

Reviewer 1 Report

Although many doctors on a daily basis, from dermatologists to plastic surgeons, prescribe cosmetic products to complement and strengthen the medical therapy assigned for the treatment of the different clinical conditions of their patients, very few of them know adequately how a cosmetic is made.

Cosmetics are today a complex category of products regulated by very specific laws all over the world and must be made with specific and safe substances that have passed the evaluation of international scientific bodies such as the FDA in the USA and the SCCS (scientific committe on consumer safeety ) in Europe.

Their ability to penetrate the skin (percutaneous absorption) and to produce biochemical modifications in the skin are now well known and account for the safety assessment to which every cosmetic ingredient must be subjected.

In this way, cosmetics today are getting closer and closer to that concept of "cosmeceutical" that Albert Kligman first described in the 1980s: a product capable of penetrating the epidermis, biochemically active, and whose effects can be evaluated through correctly conducted scientific studies.

Among the cosmetics most used by doctors, moisturizing products play a fundamental role because they are necessary for the restoration and support of the barrier function of the epidermis, often compromised by dermatoses such as psoriasis, atopic dermatitis, or simple skin dryness as in photoaging.

This type of cosmetic, like the products dedicated to other cosmetic functions, is composed of numerous ingredients that belong to different chemical classes, with different functions ranging from the conservation of the product itself to the physiological activities that can determine once applied to the skin.

For the prescriber of cosmetics today it is necessary to be able to understand, at least in broad terms, the label of the cosmetic where in the international INCI (International Cosmetic Ingredients) language the ingredients that make up the product are listed in decreasing order. This allows us to evaluate its usefulness for its intended use and also to avoid prescribing products that contain specific allergens in the event of skin allergies.

The review "Moisturizer in patients with inflammatory skin diseases" is, in my opinion, a valid guide, simple and concise, which allows you to know what a cosmetic is made from, what are the ingredients that compose it, what are their functions and what correlations exist between these and the physio-pathological processes that cause the various dermatoses.

Author Response

Thank you for your kind review.
I hope that this review paper will be of great help in the selection and treatment of moisturizers by medical staff.

Reviewer 2 Report

The article is a collection of informations about moisturizers but whitout a clear logic.

-The concept of cosmeceutical is very interesting but it is not clear why authors decide to focus on this at the beginning of the introduction and why this concept was used to introduce moisturizers. please clarify.

- Lack of integration, paragraph fragmentation, and inadequate/sloppy discussion strongly reduce the readability. Crafting clear and persuasive manuscripts is required.

Occlusive agents, emollients...etc are subtypes of mostuirizers. This has to be reflected in tha subparagraphing 

-Also. the title of paragraph "Types of moisturizers" is not so clear. Pleas clarify the concept. 

- The order of paragraphs is not so usefull to make article readable and understandable. Too many fragmentation

Author Response

Thank you for your review.
1. among cosmeceuticals, it is believed that moisturizing agents can serve as adjuvant treatments for atopic dermatitis, contact dermatitis, and psoriasis. Therefore, it is thought that doctors and medical experts can help in the treatment and research of skin diseases through a comprehensive review of moisturizers. It has been added to the introduction.
2. The subparagraphs have been clarified, and inappropriately attached explanations have been removed. This content has been reflected in the text.
3. "Types of Moisturizers" changed to "Classification of Moisturizers"
4. The order of paragraphs has been changed and subcategories have been rearranged for better understanding.

Thank you. 

Round 2

Reviewer 2 Report

Accept in present form

This manuscript is a resubmission of an earlier submission. The following is a list of the peer review reports and author responses from that submission.

Round 1

Reviewer 1 Report

This is a well-written and comprehensive (but a bit superficial) review article in terms of moisturizers. However, some sentences lack appropriate references.

Comments

# The main story is on moisturizing agents. The title may be re-thinkable, like “Cosmeceutical moisturizers for ------”

# P2, “the lipid component is supplied to the keratinocytes of the epidermis to rearrange the intercellu-lar lipids,”  Please cite suitable references.

# P2, “Appropriate application of a moisturizer not only improves the dryness, but also improves the skin's natural barrier function to protect the skin from internal and exter-nal irritants to keep the skin healthy.”  Please cite suitable references.

# P4,  “and is 170 times more effective in inhibiting moisture loss than olive oil, a vegetable oil.”  Please cite suitable references.

#P4, Typo error; “noncomedogenic and noacnegenic properties [20].” nonacnegenic?

   What is the difference between noncomedogenic and nonacnegenic?

#P6, “When the humidity in the atmosphere is less than 80%, it mainly func-tions to attract moisture below the stratum corneum.” Please cite suitable references.

#P6-7, “After topical application, it increases the moisture content inside and outside the keratinocytes, prevents the lamellar structure of intercellular lipids from being transformed from plate to crystal.” Please cite suitable references.

#P7, “PCA, as a component of natural moisturizing factor (12%), in cosmetics, has a moisturizing effect at a concentration of 3-5%.”  Please cite suitable references.

#P7, “propylene glycol has both a wetting and oc-clusive function, and at a high concentration of 40% or more, it also has the effect of dis-solving dead skin cells. “   Please cite suitable references.

#P8, “An emulsion is a two-phase system made up of two immiscible components in which the dispersed phase is contained in the form of droplets within a continuous phase. At this time, one phase is oil (oil phase) and the other phase is water (water phase).” Please cite suitable references.

#P8, “In general, moisturizers with low moisture such as ointments and oil-based creams do not require preservatives, but lotions, o/w creams, and gels contain a large amount of moisture so if they are contaminated by bacteria or fungi, there is a high probability of microbial growth, so it is necessary to use preservatives.”  Please cite suitable references.

#P10, “Healthy skin contains about 20-35% water in the stratum corneum, but when the skin barrier structure is damaged, dry skin occurs in which the water-holding capacity of the stratum corneum is reduced.”   Please cite suitable references.

#P11, “An important factor controlling the exfoli-ation of the stratum corneum is the degree of activation of various proteolytic enzymes, which are controlled by the pH and hydration level of the skin.”  Please cite suitable references.

#P11, “It is recommended that dry skin be cleansed and showered with lukewarm water and a synthetic cleanser (Syndet) made of mild surfactant be used that does less damage to skin lipids and natural moisturizers, and that users refrain from using regular soaps. “  Please cite suitable references.

#P13, “The skin of an acne patient is known to have impaired skin barrier function. In many studies, sebum secretion is increased due to increased epidermal moisture loss in acne skin, and the amount of sphingosin or total ceramide is reduced in the skin's keratin layer.” 

Please cite suitable references.

#P13, “acne patients often suffer skin irritation due to the application of benzoyl peroxide or local retinoids, resulting in increased epidermal moisture loss and severe exfoliation.” 

Please cite suitable references.

#P14, “The occurrence of comedones due to cosmetic ingredients is caused by clogged pores and appears several weeks to several months after using certain cosmetics.”  Please cite suitable references.

#P14, “acne induced by moisturizers is thought to be caused by irritation of pores, and clinically, it shows aggravation of inflammatory acne, erythe-ma, and pustules, and appears within 2-3 days after using the moisturizer.” Please cite suitable references.

#P15, “The skin of a patient with rosacea exhibits the characteristics of sensitive skin with impaired barrier function and a low threshold to stimuli such as heat or acid.”  Please cite suitable references.

Reviewer 2 Report

The flow of the manuscript is illogical and confusing.

Scientific evidence in this manuscript is insufficient to prove the beneficial effects of cosmetics for dermatological diseases.